# Factors Influencing Asia-Pacific Countries’ Success Level in Curbing COVID-19: A Review Using a Social–Ecological System (SES) Framework

**DOI:** 10.3390/ijerph18041704

**Published:** 2021-02-10

**Authors:** Gabriel Hoh Teck Ling, Nur Amiera binti Md Suhud, Pau Chung Leng, Lee Bak Yeo, Chin Tiong Cheng, Mohd Hamdan Haji Ahmad, Ak Mohd Rafiq Ak Matusin

**Affiliations:** 1Faculty of Built Environment and Surveying, Universiti Teknologi Malaysia, Skudai 81310, Malaysia; amiera@graduate.utm.my (N.A.b.M.S.); pcleng2@utm.my (P.C.L.); b-hamdan@utm.my (M.H.H.A.); akmohdrafiq@utm.my (A.M.R.A.M.); 2Tunku Abdul Rahman University College, Kuala Lumpur 53300, Malaysia; yeolb@tarc.edu.my (L.B.Y.); chengct@tarc.edu.my (C.T.C.); 3Faculty of Architecture and Ekistics, Universiti Malaysia Kelantan, Bachok 16300, Malaysia

**Keywords:** coronavirus, COVID-19, SES framework, institutional–social–ecological system, design principles, Asia-Pacific

## Abstract

Little attention has been paid to the impacts of institutional–human–environment dimensions on the outcome of Coronavirus disease 2019 (COVID-19) abatement. Through the diagnostic social–ecological system (SES) framework, this review paper aimed to investigate what and how the multifaceted social, physical, and governance factors affected the success level of seven selected Asia-Pacific countries (namely, South Korea, Japan, Malaysia, Singapore, Vietnam, Indonesia, and New Zealand) in combatting COVID-19. Drawing on statistical data from the Our World In Data website, we measured the COVID-19 severity or abatement success level of the countries on the basis of cumulative positive cases, average daily cases, and mortality rates for the period of 1 February 2020 to 30 June 2020. A qualitative content analysis using three codes, i.e., present (P), partially present (PP), and absent (A) for each SES attribute, as well as score calculation and rank ordering for government response effectiveness and the abatement success level across the countries, was undertaken. Not only did the standard coding process ensure data comparability but the data were deemed substantially reliable with Cohen’s kappa of 0.76. Among 13 attributes of the SES factors, high facility adequacy, comprehensive COVID-19 testing policies, strict lockdown measures, imposition of penalty, and the high trust level towards the government seemed to be significant in determining the COVID-19 severity in a country. The results show that Vietnam (ranked first) and New Zealand (ranked second), with a high presence of attributes/design principles contributing to high-level government stringency and health and containment indices, successfully controlled the virus, while Indonesia (ranked seventh) and Japan (ranked sixth), associated with the low presence of design principles, were deemed least successful. Two lessons can be drawn: (i) having high number of P for SES attributes does not always mean a panacea for the pandemic; however, it would be detrimental to a country if it lacked them severely, and (ii) some attributes (mostly from the governance factor) may carry higher weightage towards explaining the success level. This comparative study providing an overview of critical SES attributes in relation to COVID-19 offers novel policy insights, thus helping policymakers devise more strategic, coordinated measures, particularly for effective country preparedness and response in addressing the current and the future health crisis.

## 1. Introduction

Coronavirus disease 2019 (COVID-19) was unknown to the masses prior to its first case in December 2019 [1,2], although there were speculations that the earliest case dated back to November 2019 [3]. It was first regarded as viral pneumonia but was then analyzed to be a viral infection that has the capability to be transmitted through human-to-human interaction [4].

As of June 2020, there were countries that had successfully mitigated the problem, while some were struggling and had failed due to countless reasons. Much emphasis has been focused on the science and pharmaceutical dimensions in effort to produce vaccines to cure the pandemic, although it is believed that social behaviors and other potential non-pharmaceutical approaches, despite being intangible, indirect, and leaning towards prosociality (i.e., collective interest) via a cooperative action, can also contribute to curbing the pandemic [5]. Utilizing this research, we carried out an integration between health and the systemic institutional–social–ecological system (SES) framework, apart from knowing its relevancy when applied outside its commons domain, in order to specifically identify answers to the following research question: How can the diagnostic Institutional Analysis and Development (IAD)–SES framework help answer the level of success or failure of countries in curbing the pandemic? It has come to attention that there is a noticeable gap of research in exploring what and how institutional–social–ecological dimensions of the Institutional Analysis and Development (IAD) or SES framework help in understanding the unprecedented crisis. More precisely, given the numerous exogenous factors, this study identifies which SES attributes (design principles) are significant and effective in explaining and thus curbing the pandemic. The framework, synonymous with commons and resource management or collective action, is rarely applied to a health and disease-related topic such as COVID-19; therefore, relationships between the transmission risk or severity of COVID-19 and socio-economic and environmental factors via the SES lens should be explored in order to devise a holistic strategy in mitigating the pandemic.

The Institutional Analysis and Development (IAD) framework and SES are both developed by Elinor Ostrom [6]. The former, mostly adopted by social scientists, is primarily used to evaluate the effects of institutional arrangements, as well as the physical environment and local community in explaining a certain contextual outcome. It was envisioned as a systematic tool for scholars of different disciples to communicate with each other regardless of their broad perspectives to pave a way towards better understanding of a situation [7]. Within the framework, it is important to understand the action situation/interaction that actors are in, plausible choices made by them, and how it will affect the pattern of the outcome. In order to predict choices that will be made, we must know (i) the resources brought into the situation; (ii) the valuation assigned by the actor to the state of the world and to the action; (iii) how the actor acquires, processes, retains, and uses the knowledge and information; and (iv) the ways used by the actor in selecting a particular decision [8].

The IAD framework, however, is criticized as it lacks in terms of diversity and complexity of natural system and processes [7]. Therefore, SES is an improved version building upon the IAD framework by expanding the basic variables into more relevant categories, as shown in Figure 1. The SES framework provides a more detailed variable oriented analysis of the social–ecological system [7]. According to Partelow, SES is “*a conceptual framework providing a list of variables that may be interacting and affecting outcomes in social-ecological systems*” [9]. Via the identified SES attributes, consistent with the application spirit of IAD, we are able to diagnose and explain the interaction (activities) and the outcome of a situation. The community and governance attributes in the IAD context have been maintained as social and governance systems in an SES, while biophysical/ecological attributes converge into two sections, namely, resource systems and resource units. SES, normally applied in the context of commons governance [10], is proven to be versatile and adaptable due to its generality (see the complex adaptive system [11]). It also enables a comparison of different settings in a study where data are collected with the means to compare [9].

By adopting the SES framework, researchers have an organized variable-oriented and process-oriented line of arguments, involving systematic networks of action situations and hence a more informed decision [7,13]. Based on the research conducted by Raboisson and Lhermie [14] and Wilcox et al. [15] situating the SES framework in the human wellbeing and healthcare setting, when adapted into the context of COVID-19, the institutional–social–physical attributes (e.g., penalty, lockdown, facilities, technology, population density, and past knowledge) and their interaction (e.g., monitoring, communication, swab testing, containment enforcement) would influence the number of cumulative cases, average daily cases, and the mortality rate, and thus translate the success level of a country into low, medium, and high, as diagrammatized in Figure 2.

Theoretically, some design principles (DP) or critical success factors that are present in a successful resource management system should also exist in successful pandemic containment and mitigation. The hypothesis was that the higher the amount/frequency of design principles (successful attributes) that are present in countries, the higher the success level of them in curbing the pandemic. Despite all these factors, research focusing on socio-economic and socio-ecological drivers of the COVID-19 transmission remains scarce, when in fact it holds an equally important role as a critical determinant of COVID-19 transmission risks. Although few studies link both socio-ecological and climate factors with transmission levels of COVID-19, one study proved that the latter (climatic factor) does not have a significant effect, and hence its absence in this paper [17]. Therefore, this study attempts to contribute theoretically and methodologically by expanding the application of the SES framework and its institutional–social–ecological design principles in the COVID-19 context. Studying these factors should also be of practical significance because the findings can serve as a reference that helps provide insightful directions about COVID-19 abatement. The following sections of the paper further elaborate the topic in terms of (i) study areas; (ii) methods of data collection and analysis; (iii) results, findings, and discussions; and (iv) lastly a conclusion.

## 2. Methodology

### 2.1. Study Areas

Via the above literature review of SES design principles in terms of primary attributes and their sub-attributes in relation to the COVID-19 context, we carried out a comparative study for the selected countries located in the Asia-Pacific region. This study enlisted Japan, South Korea, Malaysia, Singapore, Vietnam, Indonesia, and New Zealand as study areas and classified them into 3 categories, namely, countries that (i) had successfully flattened the curve (i.e., high success level of abatement), (ii) attempted to flatten the curve (medium success level of abatement), and (iii) failed to flatten the curve. Such categorization was based on their cumulative cases, average daily cases, and mortality rates. Table 1 illustrates overall data of the 3 criteria for each country for the period between 1 February and 30 June 2020. However, since the dates of the data are varied as displayed in the Our World In Data website (due to the fact that each country’s first confirmed and mortality cases were different), for better data consistency and accuracy, we processed (standardized) the data prior to using them for the analysis. For example, since the specific timeframe focused here was from 1 February to 30 June 2020, cases displayed in terms of cumulative cases, daily confirmed cases, and mortality rate prior to 1 February 2020 were discounted or excluded from this study. This was important in order to reduce data distortion that would cause imprecision of the result later. From the dataset, Vietnam fared the best, with average daily cases of 2.34 and a mortality rate of 0%, as illustrated in Table 1. Figure 3 shows that the linear cumulative cases of Vietnam were also the lowest, i.e., 353. The government’s swift and strict measures had proven to be successful as the country was one of the earliest to eliminate the virus [18]. Meanwhile, Indonesia recorded the linear cumulative cases of 56,385 with average daily cases of 466, and this figure was the highest among the 7 countries. However, its mortality rate came second after Japan. See Figure 4 and Figure 5 for graphs showing daily confirmed cases and mortality rates across the 7 countries during the period. The next paragraph provides a brief background for each country as to how and when COVID-19 cases started to emerge.

Cases of COVID-19 had first emerged in these countries from early January to the end of February. In some countries, COVID-19 was brought in by Chinese nationals or individuals returning from China or Iran, such as in Singapore, New Zealand, and Japan [19,20,21]. Indonesia recorded its first case after a national came in contact with an infected person in Malaysia [22]. Even so, Japan was the first country recording a human-to-human transmission as its first case did not visit the market but was in close contact with a pneumonia patient [20]. In some countries, the first case did not necessarily lead to a booming number of cases. In Malaysia, cases started to skyrocket due to a religious event in Sri Petaling in March, attended by almost 16,000 individuals from Malaysia and other countries [23]. The same phenomenon occurred in South Korea when an elderly woman in Daegu was tested positive for the virus. She attended a 10,000-member mass gathering at the Shincheonji Church [24].

### 2.2. Data Collection

To execute a comparative analysis across the 7 countries, we had outlined a total of 13 attributes (with 5 social attributes, 3 physical attributes, and 5 institutional attributes) and their respective sub-attributes, on the basis of the 3 primary SES factors. More precisely, the 13 exogenous factors were derived from (i) the established literature of the SES framework in commons management (e.g., types of local leadership, homogeneity level, resources/technology or facility availability) and (ii) the mainstream and relevant literature and findings of COVID in terms of potential factors that affect the pandemic. Moreover, sufficient availability of the above data or information particularly pertaining to factors influencing the pandemic was also taken into account. At the same time, the interaction arena attributes comprising the government stringency index, the health and containment index, and the economic index, which are influenced by the 13 attributes, were selected as they represent the activity levels or response effectiveness of governments that eventually determine the COVID-19 outcome. To summarize this, Table 2 illustrates an operational framework that classifies 3 SES factors that comprise attributes and sub-attributes (such as exogenous variables) and their working definitions, components within the interaction arena (activities/responses), and the success level of COVID-19 abatement (as an outcome variable).

This study made use solely of secondary sources, such as research papers, government documents/policies, official newsletters, and websites specifically pertaining to gathering COVID-19 data on the basis of the predetermined SES attributes across 7 countries; statistical data in terms of the number of COVID cases; and COVID-related research findings on factors influencing COVID-19 severity or transmission risks as well as governments’ responses when dealing with the pandemic. Keywords such as “IAD-SES framework”, “social–ecological system”, and “COVID-19” were used on Google Scholar to garner information. Other than this, we also referred to ourworldindata.org to collect statistical data on cumulative cases, mortality rates, and the average daily confirmed cases. Moreover, prior to explaining the COVID-19 (abatement success level) outcome of a country, on the basis of the Oxford COVID-19 Government Response Tracker (OxCGRT) numerical data, we also referred to the 18 policy indicators of government responses categorized into 3 main indices, namely, the Government Response Stringency Index (on containment and closure policies, such as school closures and restrictions in movement), the Health and Containment Index (on COVID-19 testing regime or emergency investments into healthcare), and the Economic Index (on income support to citizens or provision of foreign aid), as they served as an important data input for the “interaction or action arena” (responses) of the SES framework to ensure the coherency and validity of the data (i.e., attribute input-influencing factors). However, it is worth noting that the data extracted only cover information from the month of February 2020 to June 2020. The specific timeframe was chosen as most country cases were at their peak.

### 2.3. Data Analysis

The study primarily employed a qualitative content analysis to study institutional–social–ecological attributes using Ostrom’s 8 design principles [6] as one of the theoretical underpinnings. To aid the comparative analysis, we constructed a coding system, based on both textual and quantitative evidence, and used it to determine IAD-based SES attribute occurrence in each country, meaning that each attribute was assigned with a code on the basis of the ratio or percentage of DP/attribute presence/occurrence [45], i.e., mostly absent/absent (A) (with 0–29%), sometimes/partially present (PP) (with 30–69%), or mostly present or present (P) (with 70–100%). This coding process, particularly for the comparative analysis of a limited number of cases where a statistical inferential analysis is not possible, is scientifically appropriate and relevant since it is based on several SES scholars’ systematic methodologies that have been widely applied in different resource management contexts [46,47,48]. Ultimately, in line with Ostrom’s successful commons governance focusing on the importance of the presence of design principles, a higher frequency/co-occurrence of P indicates a better COVID-19 containment and therefore a higher success rate of the country, while a lower count of P connoting a higher frequency of A or PP means otherwise. Moreover, a quantitative method, apart from providing the percentage of DP occurrence and assessing the frequency of attributes present as well as for the purpose of calculating scores and assigning ranks was also used to supplement the qualitative analysis. To ensure the consistency or credibility of the coding process between two coders, we conducted inter-rater reliability of Cohen’s kappa using SPSS for the reliability of all 12 SES attributes (excluding effective foreign worker containment) across 7 countries where, instead of seven items per attribute which are inadequate in terms of the minimum sample size, altogether 84 items were analyzed. The cut-off point for substantial or good reliability/agreement is 0.61-0.80 [49,50]. Next, the 3 indices that form the interaction arena (government responses), in terms of the scores, were assigned with different codes, namely, high (H), medium (M), and low (L) (see Table 3). Since the data for each index varied from time to time, the highest score was considered when assigning the code, instead of an average score; considering the countries’ different COVID-19 severity levels and situations that affect the indices, using the latter may not be accurate as it probably understates the true level or index of a country.

At the same time, the success level of a country (outcome), influenced by the above DPs and 3 indices, was measured and determined on the basis of the total scores obtained from the 3 criteria, namely, the cumulative number of cases of each country, average daily confirmed cases, and the mortality rate of the country [17,44]. More precisely, the total scores were calculated based on the ranking among the 7 countries (i.e., first to seventh) and values assigned (from 1 to 7) to each criterion. Table 4 summarizes the options of values for each rank as well as the score ranges (from 3 to 21) for the high, medium, and low outcomes.

## 3. Results and Discussions

On the basis of the analysis, we show in Table 5 a summarized result spanning seven countries with respective SES attributes and interactions, as well as the success level of abatement. On the whole, with the overall kappa reliability of 0.76 where *p* < 0.01, the coding of 84 items under the 12 SES attributes was considered substantially reliable. It was found that in terms of the ranking and scoring values, only Vietnam and New Zealand, with scores ≥ 16, were deemed successful or had high success in terms of COVID-19 abatement, while Japan and Indonesia were deemed the least successful countries in abating the pandemic, with scores less than 10. The remaining three countries, namely, Malaysia, Singapore, and South Korea, were considered to be at the medium level. For the presence or occurrence level/frequency of SES attributes, New Zealand and Singapore had the highest number of Ps, (i.e., eight Ps and one A), followed by Vietnam (with seven Ps and one A) and South Korea (with six Ps and one A). While Malaysia had scored the most PPs (i.e., eight), Indonesia and Japan had the highest number of As, i.e., at least two As for the SES attributes. From this result, we found that the SES design principle/attribute presence level appeared to be associated with the three indices on the government activity and response level, which therefore help explain the success level of COVID-19 abatement. Generally, countries with a higher number of Ps for SES attributes were more likely to score at a medium or high level for the overall three indices (e.g., New Zealand, South Korea, Singapore), except for Vietnam, with the high number of Ps having scored a low level for the economic index. The prevalence of association between attribute occurrence and the index level was even more prominent for countries such as Japan and Indonesia that had the lowest number of Ps or highest number of As for the SES attributes, where this tended to be associated with the low-level indices. The level of indices (whether it was high, medium, or low) reasonably determined by the co-occurrence of SES attributes consequently explained the abatement success level of a country in terms of the number of infections (daily average and cumulative) as well as mortality rate; the higher the level of indices, particularly for the Government Stringency Index and the Health And Containment Index, the higher the success level of a country in abating the pandemic (see Vietnam and New Zealand) and vice versa. More detailed explanations of results for each country in terms of each SES attribute are provided later.

### 3.1. Low Population Density

In a study performed by Kadi and Khelfaoui in Algeria, the authors found that population density indeed played a role in the transmission of the virus [51]. This is also supported by a study done by Bhadra et al. in different districts of India [26]. However, it cannot be a strong explanatory factor as to why the number of cases are as known [52]. Among the seven countries, based on the rank-size rule, distributing population density into 3 categories, Singapore, coded A, has the highest population density (8358 per km^2^), as shown in Table 5. Despite the high population density, the country, having adequate facilities and economic strength, managed to keep the number of positive cases and the death rate under control. In contrast, Indonesia fumbled, despite having a lower population density; this is probably due to other SES factors, such as shortage of healthcare facilities, weak institutional arrangement for strict enforcement, and also economic conditions as one of the considerations.

### 3.2. High Social Homogeneity

As seen in Table 5 above, Vietnam, South Korea, and Japan were justified as P because their populations appear to be relatively homogenous in terms of culture and some demographic attributes (e.g., ethnics and language). Meanwhile, other countries were considered as PP due to their heterogeneity in terms of multicultural and racial settings, such as Malaysia, Indonesia, New Zealand, and Singapore. Culturally, Japanese people have always upheld a high standard of hygiene and obedience (compliance). They have always practiced social distancing as they respect personal space. Wearing masks when feeling sick is a norm to prevent passing a virus to others [53]. Hence, there are no objections among its citizens when mask rules were enforced by the government. However, to posit the idea that the homogeneity of a nation contributes positively to the rate of success seems rather farfetched as the analysis showed that it does not play a major role.

### 3.3. High Level of Trust

Table 6 shows the level of trust among citizens towards governments of each country. Having trust from the people ensures that the public follows the rules undisputedly (i.e., higher compliance), allowing the leaders to convince them of mass testing and quarantines before things become worse and hence keeping the virus at bay earlier on [29]. In other words, having citizens’ high trust towards the government may contribute positively to the success level of a country in curbing the pandemic. For example, countries such as Singapore and New Zealand are deemed to have trust due to transparency and the leader’s excellent risk communication. Meanwhile, other countries’ governments (Malaysia, South Korea, and Vietnam) had constantly proven their competency, hence garnering public trust in their capabilities [24,29,31,54,55]. In Vietnam, the people believe that the government is working for the betterment of the country; the Vietnamese government is also competent in providing prompt communication and medical supplies, thus allowing them to carry out strategies similar to wealthier countries despite being a lower income nation [29].

Scoring 0.486 in a public survey conducted by Fetzer et al. [56], the Republic of Indonesia has been speculated to be hiding the real number of cases in the country by activists and political opponents. Japan’s government too faced the same problem. The government has been labelled distrustful by the Edelman Barometer, and this was also backed up by the public survey, where the survey stated that more than half of its citizens (60.2%) had perceived the government as being untruthful in containing the spread of COVID-19 [56]. To date, all countries practice a coordinated information campaign, which promotes effective communication between the government and citizens, allowing a two-way communication between both sides. This ensures citizens are well informed on the current state of virus spread in the country and ways to mitigate it. In certain country, this step helps solidify the government as a prominent leadership figure in the country.

### 3.4. Sufficient Local Management Knowledge and Experience

Sufficient local management knowledge and experience are mostly present in all seven countries, as seen in Table 5. These attributes take into account each country’s experience in dealing with similar diseases to COVID-19, which in this case are Severe Acute Respiratory Syndrome (SARS), Middle East Respiratory Syndrome (MERS), and Novel Influenza A (H1N1). The outbreak of SARS, H1N1, and MERS occurred in 2003, 2009, and 2012, respectively. SARS, MERS and COVID-19 are caused by *Betacoronavirus* genus while Influenza A viruses lead to H1N1. SARS and MERS possessed a lower transmission risk but significantly higher fatality rates [57]. As for H1N1, it has been classified as having a self-limited mild-to-moderate transmission risk, although there have been reports of fatal outcomes [58].

These experiences helped certain countries better prepare and improve a healthcare system in infection prevention and control, activate response protocols, increase number of thermal scanners at all borders, and realize the importance of isolating infected cases and quarantine measures [54,59,60]. This means that countries with the past experiences and knowledge in handling related pandemic contagions can cope better with COVID-19. Contrary to other countries (e.g., New Zealand, South Korea, Vietnam, and Malaysia) who have experienced at least two pandemics, Japan has comparatively less past experiences (i.e., as seen in Table 7, only experiencing one pandemic). For example, in terms of its one-time H1N1 experience, Japan opted for less testing in handling the COVID-19 pandemic. This was probably due to the previous outbreak experienced by Japan that saw the public flocking to hospitals to get tested, which subsequently increased the transmission risks. Thus, the Ministry of Health, Labor and Welfare was concerned that crowded hospital lobbies would lead to a surge of COVID-19 cases and hence finalized their decision to limit accessibility [61]. Perhaps such a decision based on its limited experience may be rather ill-considered, since other countries that had experienced more than one pandemic took a different course of action (e.g., South Korea instead promoted more testing in order to curb the transmission risk). However, despite the above, in which the failure of Japan is well justified, Indonesia, coded PP similar to New Zealand, Vietnam, and Singapore, showed otherwise, where it also failed to control the pandemic. It can be deduced that this seemingly significant attribute may not necessarily be critical in explaining the severity of the pandemic.

### 3.5. Effective Foreign Worker Influx Containment

Table 5 shows that both Malaysia and Singapore had a large influx of foreign workers, leading to clusters of COVID-19 cases. If there is a proper containment system of those workers, it would be indicated as P. On the other hand, other countries were marked N/A, which means this attribute was not applicable as they did not have a huge influx of foreign workers leading to COVID-19 cluster emergence. A proper containment system helps eradicate possibilities of a surge of cases, therefore controlling the number of cumulative cases and maintaining a low mortality rate.

The rise of a foreign worker cluster in Singapore was apparent from 15 to 30 March, comprising almost 61% imported cases. Both Malaysia and Singapore cited that poor living conditions had sparked a widespread infection among the group [63,64]. The severity of cases in Malaysia forced the government to employ a more stringent lockdown, called the Enhanced Movement Control Order (EMCO) [65].

### 3.6. High Facility Adequacy

In terms of determining the code in Table 5, P translates to countries with high healthcare facility adequacy, while PP translates to moderate adequacy, and A translates to a low facility adequacy. Having an ample amount of resources (physicians and facilities) translates to a better mobility, hence allowing more patients to be treated at one time. This would ensure better healthcare services to all. Therefore, the COVID-19 mortality rate could be lowered. Generally, high-income countries such as South Korea and Japan possessed sufficient health facilities and manpower (Table 8), except for New Zealand and Singapore that had a lower number in terms of healthcare facilities [66,67]. However, the lower income country like Vietnam managed to control the pandemic, probably due to its governments’ swift strategy as they recognized its inability to contain a high amount of cases if the pandemic became serious [68].

Indonesia has the lowest density/number of physicians among the seven countries. In addition, its facilities are in a dire condition [71]. The country’s Ministry of Health has recognized an uneven distribution of health facilities and quality across Indonesia [71,72]. Some sub-districts in Indonesia did not possess any health center and lacked necessities such as electricity, clean water, and proper equipment, with limited transportation [73]. An obvious disparity between rural areas and urban regions is common in many countries, including Malaysia. Facilities in these rural areas are usually government-funded with fewer doctors and hence a lower doctor-to-patient ratio [74].

### 3.7. High Technology Availability

The presence of technology aids governments and healthcare sectors in devising strategies towards the abatement of the virus. As seen in Table 5, countries with code P have technologically advanced medical equipment production and have also developed their own test kits. Having PP for this attribute means limited medical equipment and lacking in test kit production. All countries had also adopted the use of a contact tracing application (see Table 9). In Malaysia, the efficiency of contact tracing applications is moderate, as a study showed that the application MySejahtera uncovered 251 confirmed COVID-19 cases or 3% of 8308 cases as of June [75].

Some countries had also developed their very own diagnostic kit that can detect COVID-19 to bolster testing rates in their respective country [26,50,85]. Indonesia’s kits could be mass-produced with a lower budget, hence lowering expenditure [80]. Next, drive-thru testing has proven that this system is efficient, as South Korea recorded a higher testing number due to 50 drive-thru testing facilities being set up as of April 20 [54]. In addition, most countries produce their own personal protective equipment (PPE) and ventilators as global supplies dwindle and most manufacturing companies focused on meeting their own country’s needs [92]. The amount of medical equipment should be proportional to the country needs to ensure all patients were given the chance to be treated and healed, thus decreasing mortality rate of the country.

### 3.8. Economic Performance

As seen in Table 10, New Zealand, Singapore, South Korea, and Japan were categorized as high-income countries. Meanwhile, Malaysia and Indonesia were grouped as upper middle-income and Vietnam as a lower middle-income country. Having immense resources based on the economic performance/status allows the government to invest in better facilities, technology, and remuneration packages that cover broad relief during the crisis, as seen in Singapore [31]. In addition, this may allow the government to exercise stringent rules and have effective enforcement. To date, Indonesia has the highest percentage of poverty among the seven countries, with half of its population struggling to make ends meet. A national lockdown may not be a preferred choice as it causes their economy to collapse, affecting their livelihood and putting tens of millions of its people who worked poorly paid informal jobs at risk [93,94]. Interestingly, despite being a lower middle-income country, which possibly undermines COVID-19 containment strategies, Vietnam, with the lowest GNI per capita rank i.e., 141^th^, managed to curb the pandemic successfully, with this potentially suggesting that the economic enabling factor may not have direct, significant impacts on COVID-19 outcome.

### 3.9. Top-Down Leadership

During the pandemic, the political system of each country plays a vital role in maneuvering the nation through the pandemic. Table 11 further elaborates on Table 5, explaining the local leadership status where P, PP, and A translate to authoritarian, flawed democracy, and full democracy, respectively. These labels were derived from the Economist Intelligence Unit Democracy Index, which takes into account government corruption, popular perception of citizens’ freedom of choice, public confidence in the government, and the government’s extent of jurisdiction [34].

In Vietnam, the role of the government is very apparent as they were authoritarian in handling the pandemic. The government is more aggressive in enforcing laws without any delay and public debate [98]. The Vietnamese government had no problem mobilizing their military for healthcare missions while enforcing strict restrictions publicly. In an effort to ensure all citizens follow the social distancing and quarantine rule, Vietnam’s entrenched system of loyal neighborhood party cadres took the liberty to spy on residents and to report their sightings to the superior [68]. Such an authoritarian approach, also practiced in China, has been praised by the WHO’s director general, although it seems to contravene the organization’s constitution with respect to the “*fundamental rights of every human being without distinction*” [99]. However, New Zealand, with the Democracy Index score of full democracy, showcased excellent COVID-19 abatement. Although Kleinfield argues that types of regime do affect a country’s success, this alone does not sufficiently contribute to explaining the outcome [29], and the New Zealand full democracy case is definitely worth revisiting since many scholars discovered that countries that are pro-authoritarian can control the pandemic better.

### 3.10. Penalty

Table 12 outlines the compounds imposed by the seven countries for potential offenses committed during the pandemic period. Most of the time, the amount of penalty is high to instill fear and compliance among the public. Thus, people are more inclined to adhere to the outlined Standard of Procedures (SOPs), consequently curbing the virus transmission risks. As opposed to other countries, Japan did not impose any penalty on its citizens, as summarized in Table 5. Instead, it relied heavily on the people’s obedience [61,100]. The Japanese law prohibited the government from enforcing stringent penalties as it was formed on the basis of the idea that human rights should be respected. This can be linked with incidents during World War Two, wherein Japan had memories of civil rights abuse [101]. Therefore, the government can only “request” the people to minimize their movement. Social pressure is at work as the public averts risk of bearing responsibility for spreading the virus as they fear social sanctions [61].

### 3.11. Strict Lockdown Enforcement

As cases spike, many countries resort to the implementation of lockdown to contain the virus from spreading. Table 13 shows that although most countries employed a strict national lockdown together with public area closures (P), only some endorsed closure of public areas and gathering bans (PP) while others practiced little to none of them (A). Lockdown becomes more effective especially when paired up with policies of social distancing and mandatory mask wearing. It has been proven to be a great circuit breaker as it restricts movement and ensures safe proximity among citizens. Therefore, possibility of a human transmission would be low, leading to lowering cumulative cases, daily case increase, and death tolls.

In Japan, the government is incapable of enforcing lockdown due to the Japanese Law [53]. However, an amendment was made to allow the prime minister to declare a “state of emergency” in areas that are heavily inflicted by the virus. In addition, although the government had called for remote working, most companies are not equipped and prepared for such a scheme [53]. Both South Korea and Indonesia applied a national social distancing policy instead of a national lockdown (Table 13). Localized lockdowns are to only be implemented in highly infected areas [94,123]. However, the amount of comprehensive rapid testing performed by South Korea set their outcomes apart. Likely due to the poorer state of the economy, Indonesia’s response was slow and unclear, which in the end cost the country a spike in cases.

### 3.12. Strict Standard of Procedure in Public Areas

Table 5 shows that most countries were coded P, with the implementation of all SOPs except for the mask-wearing requirement (see Table 14). Severe Acute Respiratory Syndrome Coronavirus 2 (SARS-CoV-2) spreads by water droplets that can be transmitted through air or on surfaces. The WHO suggested mask wearing as an integral strategy to suppress transmission coupled with other policies such as social distancing, health checks and temperature scanning in public areas, washing hands, and sterilization. Sterilization and fumigation procedures are vital in frequently used public spaces, such as public transportation stations, shops, and especially in infected venues. These SOPs are crucial as they can break the virus chain if abided accordingly, hence lowering infection cases.

Japan was assigned PP although there is implementation of SOP in public areas. This is due to the fact that these SOPs are recommended rather than enforced [53]. As stated previously, the unique Japanese law does not allow the government to impose their citizens with a strict law, instead only hoping that they will adhere to the measures.

### 3.13. Emergency Response Plans and COVID-19 Testing Policies

An emergency response plan in a pandemic situation is crucial to ensure an effective control of the pandemic. Table 5 shows that all countries had devised their own emergency response plan, although some may not be as rigorous as the others’. Among the popular forms of emergency response plan are debt relief and income support by the government, as shown in Table 15. Most countries opted for <50% lost salary coverage with a narrow relief, applied by Indonesia, Japan, South Korea, and Vietnam. The Singapore government devised a >50% salary coverage plan with a broad relief. Having a broad relief helps decrease the citizen’s movement as they can stay home without worrying much about their financial conditions. Restricting movement proves to be an excellent circuit breaker to the virus transmission.

As a step towards preparing for the worst, the governments also called for the increase of healthcare facilities by turning buildings into quarantine centers, mobilizing more labs for testing and research purposes [60,136,143] For example, Indonesia repurposed a truck into a laboratory to increase testing rates [84]. The South Korean and Malaysian healthcare systems also transformed a few hospitals into special COVID-19 hospitals [23,137]; thus, specialized healthcare workers could prioritize treating COVID-19 patients until recovery.

Moreover, COVID-19 testing policies also play a crucial role in breaking the virus chain as the infected can be promptly identified and isolated or quarantined. Countries with a more inclusive and comprehensive swab testing policy, which conducted open public/mass testing covering both symptomatic and asymptomatic testing (e.g., Vietnam, New Zealand, Malaysia, South Korea) may have a better COVID-19 abatement outcome. On the other hand, countries (e.g., Indonesia) with limited swab testing, probably due to the above economic or facility availability constraints, may have a COVID-19 outcome that will become severe since many cases, especially asymptomatic ones who are also highly infectious, go undetected.

### 3.14. Interaction Arena

In addition to the SES attributes described above, activities or government response effectiveness in terms of efficient monitoring, communication, movement restriction, and extensive contact tracing and testing were used as supplemental evidence and references to further explain the COVID-19 abatement/severity outcome of each country. There are three indices referred to, namely, the Government Response Stringency Index, the Health and Containment Index, and the Economic Support Index.

New Zealand and Vietnam recorded the highest stringency score at 96, followed by South Korea at 82, Indonesia at 80, Singapore at 76, Malaysia at 75, and Japan at 47 [18]. High stringency, composed of indicators such as school and workplace closure, public events or gathering restriction, and international and domestic travel controls, reflects governments’ strict policy emphasizing effective monitoring. In terms of the results, New Zealand and Vietnam were labelled as high, while Japan was considered low. Other countries such as Singapore, South Korea, Indonesia, and Malaysia were considered medium.

Meanwhile, Vietnam had the highest health and containment index with a score of 85, followed by New Zealand scoring 79, South Korea at 74, Singapore obtaining 73, Malaysia at 71, Indonesia at 69, and finally Japan at 41 [18], where in terms of the analysis, both Indonesia and Japan are considered low. The Containment and Health Index measures indicators such as testing policies (PCR tests) and extent of contact tracing. Each of these countries has varying testing policies ranging from only available to those with symptoms and key groups to open public testing that allows even asymptomatic patients to take part, as shown in Table 15. Despite having a policy that allows anyone to undergo swab testing, Japan has the second lowest swab testing frequency (of 7 days average per 1000) as of 30 June at 0.03, coming second after Indonesia at 0.04. Singapore has the highest average at 1.92 followed by New Zealand at 1.31, Malaysia at 0.32, South Korea at 0.22. and lastly, no data are available for Vietnam [18].

Looking into the statistics of the Economic Support Index, Singapore obtained the maximum value of 100 in providing economic support to its citizens, followed by Malaysia and Japan at 75, New Zealand at 62, South Korea at 50, and Indonesia and Vietnam at 25. The Economic Support Index illustrates the government support in terms of covering salaries, providing direct cash payments, or universal basic income or similar for the unemployed, freezing financial obligations such as loans and services [18].

Due to its financial capacity, Singapore is able to allocate a huge amount of money to ease the financial burden of its citizens. This results in having a sufficient amount of financial aid provided by the government, which lessens the need to work amidst the pandemic. Therefore, Singaporeans are able to comply to the government rules and stay home unless necessary. Thus, it is appropriate to say that the indices relate and have strong dependency with each other. This scenario differs for a country with a different economic reality. By contrast, Indonesia and Vietnam scored 25, the lowest among other countries. Indonesia, with almost 9.6% of its population living below the national poverty line, is unable to cater enough financial aid to each of its citizens. Its citizens working in informal settings with low wages are left with no choice but to work, despite stay-at-home restrictions [93,94].

Although financial consideration does play a big part in determining the COVID-19 abatement level of a country, it is not a major deciding factor. As seen in Table 10, Vietnam does not have an economy as robust as Singapore and yet the former managed to mitigate the virus earlier than its counterpart due to its efficient government response policy [68]. Vietnam, scoring a high tier in both the Government Stringency Index and the Health and Containment Index entails its government with strong leadership presence, with strict monitoring and efficient communication skill, as well as a comprehensive contact tracing policy [51,111,131]. Hence, citizens are well informed and possible cases can be mitigated before emerging into a deadly cluster. Despite having the lowest score among the other countries in the Economic Support Index, Vietnam is compensated by its strong government response policy. A similar scenario could be seen in New Zealand, with a financial capacity significantly lower than others. Although scoring only 62, New Zealand successfully lowered its COVID-19 cases and mortality rate approaching 30 June 2020, as seen in Table 5.

Therefore, it can be deduced that the level of COVID-19 abatement can be rather directly justified by the scores reflected in the Government Stringency Index and the Health and Containment Index. Countries scoring a high score in these indices generally have a high cumulative score, as shown in Table 5 (Vietnam and New Zealand). Meanwhile, countries with significantly lower scores in both indices had a low success rate in abating the virus. This is prominently seen in Japan, which scored low in both indices. Despite being in the medium tier in providing economic support, Japan is still struggling to control the number of COVID-19 cases in its country.

## 4. Conclusions and Recommendations

Much effort has been focused on pharmaceutical aspects in combating the COVID-19 pandemic; thus, in recognizing the gap and the need in exploring non-pharmaceutical interventions and strategies in stemming the transmission risk, this diagnostic study’s primary objective was to explain COVID-19 severity from an SES perspective, hence testing out the framework’s adaptability in the non-commons or health context. Through the SES analytical framework providing a theoretical basis, we reviewed seven Asia-Pacific countries for institutional–social–ecological factors that affect their success in fighting the pandemic. Probably due to adherence to, or the high presence (P) frequency of, the design principles, which subsequently reflect the high level of the Government Stringency Index and the Health and Containment Index, both Vietnam and New Zealand were classified as countries with the high success level of COVID-19 abatement, which is coherently explainable. Meanwhile, Japan and Indonesia, with low presence of (P) frequency or with the high presence of (A), were deemed as countries with the low success level.

However, although some countries possessed a high number of Ps for SES attributes, the outcome signified that some countries (e.g., South Korea and Singapore) with the intermediate level of success may not effectively abate the pandemic in comparison with what has been hypothesized. Thus, two lessons can be drawn: (i) having high number of Ps for SES attributes does not always mean a panacea for the pandemic; however, it would be detrimental to a country if it lacks them severely, and (ii) some attributes (mostly from the governance factor) may carry higher weightage towards explaining the success level. More precisely, the critical success factors/design principles found in Vietnam and New Zealand are ultimately about the strict implementation of lockdown, penalties, high trust level of citizens towards government, highly adequate and efficient mobilized facilities, and comprehensive COVID-19 testing policy. The economic situation of a country does play a vital role in keeping the pandemic under control, but Vietnam demonstrated otherwise or is an “anomaly” in handling their COVID-19 situation, given its status as a lower middle-income nation by the World Bank. Meanwhile, low success outcome countries, such as Japan and Indonesia, seemingly are associated with the lack of national lockdown implementation and low testing rates, and specifically in Indonesia, it failed probably due to its economic situation, inadequate facilities and resources, and manpower to contain the pandemic. Other factors, such as population density, social homogeneity, past experience and knowledge in managing an infectious disease, and local leadership regimes, are not as impactful and significant as the above factors.

By understanding the overview of SES attributes in relation to COVID-19 and identifying which are potentially the most significant ones, this comparative study, via the SES methodical coding framework, provides knowledge and best practice exchange between the seven countries, shedding light on how to leverage these critical institutional–social–ecological attributes so that more strategic policy guide and coordinated measures for effective country preparedness and response in terms of managing and addressing the current and the future health crisis can be devised.

Despite the above insights and contribution, this study with a rather small number of case studies was, however, limited to secondary data and literature reviews, which were primarily analyzed in a qualitative descriptive manner. Meaning that information and conclusion (i.e., indicative patterns of significant SES attributes for COVID-19 abatement) drawn from this study may not entirely be generalizable to other countries’ in curbing COVID-19 and do not provide absolute quantitative accuracy. Therefore, for further validation of the existing SES attributes in terms of how applicable and valid these factors are determining the COVID-19 severity level, future studies may consider extending the SES framework application to include more geographical regions and countries. In addition, focusing on the inclusion of higher-tier SES attributes and exploration of other non-pharmaceutical factors, cross-sectional studies via the public perception survey coupled with robust inferential statistical analyses for more accurate explanation and prediction of the pandemic outcome are necessary.

## Figures and Tables

**Figure 1 ijerph-18-01704-f001:**
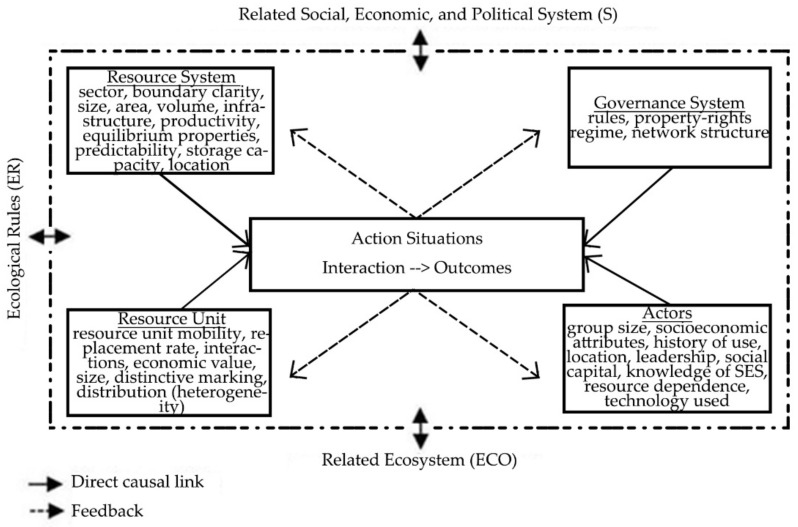
Second- and third-tier social–ecological system (SES) attributes. Adapted from Ostrom and Cox [12].

**Figure 2 ijerph-18-01704-f002:**
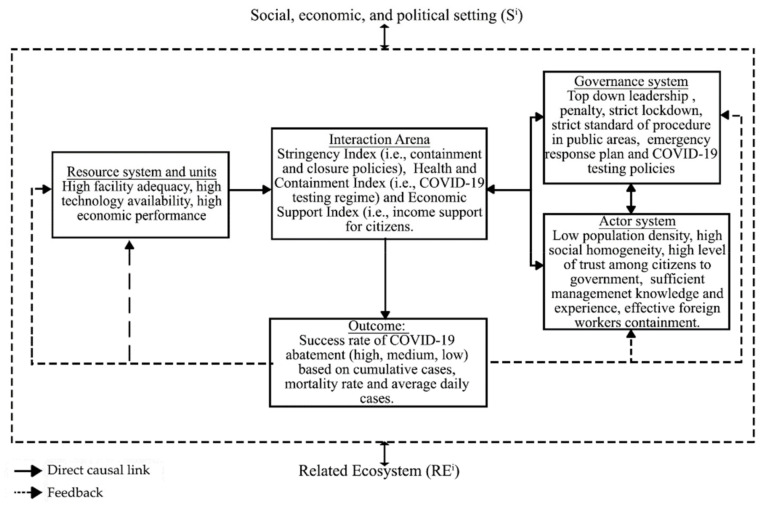
The SES framework in the Coronavirus Disease 2019 (COVID-19) context. Source: Adapted from McGinnis and Ostrom [16].

**Figure 3 ijerph-18-01704-f003:**
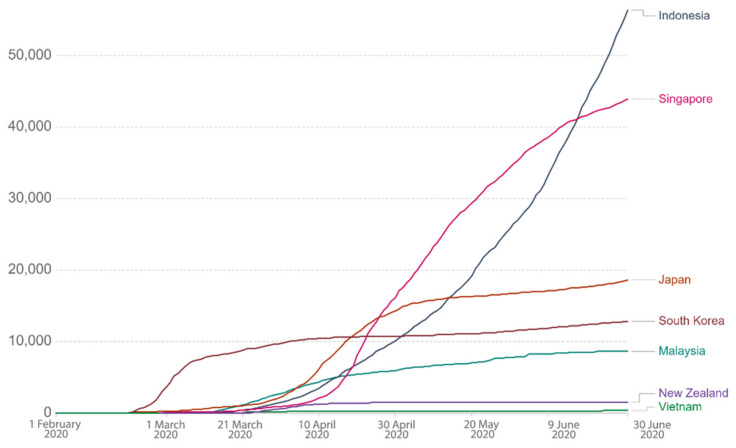
Cumulative COVID-19 cases. Source: Roser et al. [14]. Retrieved from ourworldindata.com.

**Figure 4 ijerph-18-01704-f004:**
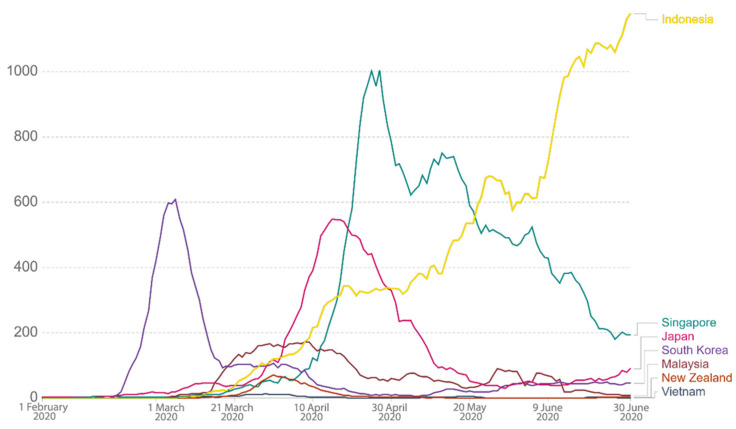
Daily confirmed cases (7 days average). Source: Roser et al. [14]. Retrieved from ourworldindata.com.

**Figure 5 ijerph-18-01704-f005:**
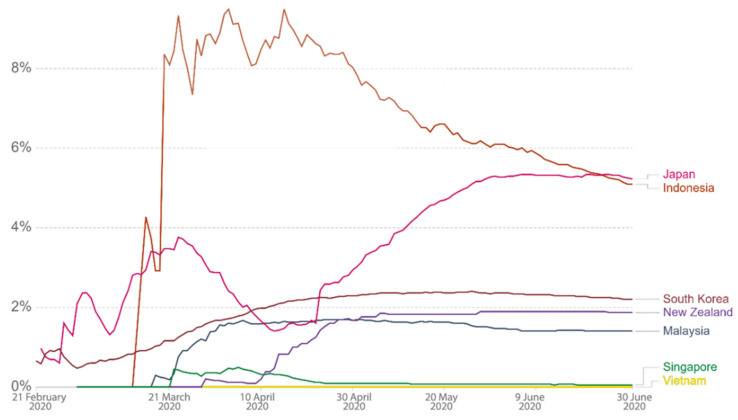
Mortality rates of each country. Source: Roser et al. [14]. Retrieved from ourworldindata.org.

**Table 1 ijerph-18-01704-t001:** Data of cumulative cases, average daily cases, and the mortality rate from 1 February to 30 June 2020.

Country	Cumulative (Total) Confirmed Cases *	Average Daily Confirmed Cases **	Mortality Rate (%) ***
Indonesia	56,385	466	5.1
Japan	18,600	123.2	5.2
Malaysia	8631	57.2	1.40
New Zealand	1528	12.3	1.44
Singapore	43,894	290.7	<0.1
South Korea	12,839	85.0	2.2
Vietnam	353	2.34	0

*,** from 1 February to 30 June 2020. *** from 20 February to 30 June 2020. Source: Adapted from Roser et al. [14] ourworldindata.org.

**Table 2 ijerph-18-01704-t002:** An operational framework comprising list of attributes, its working definitions, and effects towards COVID-19 abatement.

Factors	Attributes	Working Definitions (Sub-Attributes)	Assumptions/Effects towards COVID-19 Abatement (Outcome)
Social	Low population density	Number of population per land area/size of a country [25,26]	With lower population density, it may have a lower risk of COVID-19 transmission considering more spacious environment for less close human interaction. High population density tends to be associated with the overcrowding issue, and this may likely lead to decreases in the quality of living conditions and sanitation, and thus the transmission rate may be high.
	High social homogeneity	Homogeneity in terms of socio-cultural and demographic context (e.g., background, interest, ethnicity, and language) [27]	With higher social homogeneity, it may result in the higher cooperation level among the people and less self-interest among the community, emphasising collective interest and thus the higher level of COVID-19 abatement.
	High level of trust among citizens	Trust level between the community and governments measured by effective communication and government’s capability [28,29]	With a higher level of trust between the community and the government in terms of governments’ capability in handling the pandemic and effective communication, there may be higher compliance to government rules, and therefore better COVID-19 abatement.
	Sufficient local management knowledge and experience	Taking into account previous contagions, e.g., SARS, MERS, and H1N1 that have similar features as COVID-19 [30]	With the past experiences and knowledge in dealing with previous pandemics, a nation (the government and the people) may have better abatement strategies.
	Effective foreign worker containment	Special containment measures and environments in terms of accommodation and facilities provided for foreign workers to avoid local cluster outbreak [31]	Effective foreign worker containment via specific measures can have a lower transmission risk and thus better COVID-19 abatement.
Physical/resources	High adequacy of facilities [32]	This attribute covers the density of physician (per 10,000) and healthcare facilities	Countries with a sufficient number of physician and healthcare facilities signifying high mobility of those facilities for COVID-19 treatment can better abate the pandemic.
	High technology availability [33]	Concerning whether a country is equipped with high technological tools and facilities in dealing with COVID-19, including contact tracing application, testing policy, test kit production, ventilator production, PPE production	It is hypothesized that a country equipped with high technological facilities and tools can cope better with the pandemic in terms of providing timely and sufficient treatment for patients.
	High economic performance [31]	Considering the GNI per capita of a country, which determine the economic status by the World Bank	High-income countries that signify high economic performance in terms of GNI per capita can cope better with the pandemic by providing financial relief and more efficient technological/facility support.
Governance/institutional	Presence of top-down leadership [34]	Measured by the Democracy Index (based on the types of government leadership—democratic or authoritarian).	Countries with strong or firm (autocratic) leadership presence can better ensure compliant behaviours (i.e., better cooperation among citizens) and therefore better COVID-19 abatement.
	Strict penalty [35]	Some form of punishment or fine is imposed on several conditions, namely, fleeing quarantine, violating lockdown, incompliance with the government order, not wearing mask at public areas, and spreading fake news	Countries with penalty imposition on violators may have a better control of the pandemic. It is believed that with this stringent imposition, self-interest behaviour of an individual can be discouraged and thus prosocial (compliant) behaviour towards collective interest can be promoted, which is better for COVID-19 abatement
	Strict lockdown imposition [36,37,38,39]	Lockdown imposition covers school closure, border closure, gathering bans, work from home policy, national lockdown, localized lockdown, and military enforcement	Countries with strict lockdown enforcement limiting certain non-essential activities (lesser human interactions) can help stem local transmission risks and thus reduce daily confirmed cases or fatality rates.
	Strict standard of procedure in public areas [40,41]	Mask-wearing, social distancing, health checks and temperature scanning	Countries with strict standard of procedure imposition, especially in public areas, for promoting effective monitoring and reducing transmission risks can help curb the pandemic.
	Emergency response plan and COVID-19 testing policy [42]	This attribute emphasizes economy stimulus, repurposing of existing buildings into healthcare facilities, and the COVID-19 testing policy (i.e., whether the test is required by anyone with symptoms or without symptoms)	Countries that provide economy stimulus for its citizens for livelihood purposes can help curb the local transmission risk, and countries with a plan to repurpose some buildings into healthcare for accommodating and treating more patients can help reduce transmission risks and daily cases or mortality rate. Countries adopting a more inclusive/comprehensive COVID-19 testing policy can detect cases earlier and control the spread more effectively.
Action arena (activities/response effectiveness level for the period between 1 February and 30 June 2020)	Government Stringency Index [18,43]	This index is about containment and closure policies, such as school closures and restrictions in movement	These three indices reflecting the governments’ activities and response effectiveness, determined by the attributes define the outcome (i.e., the COVID-19 abatement level). High stringency, containment, and economic support indices are associated with the higher success level of COVID-19 abatement.
	Health and Containment Index [18,43]	This is about COVID-19 testing regime or emergency investments into healthcare	
	Economic Support Index [18,43]	This is about income support to citizens or provision of foreign aid	
Outcome	COVID-19 abatement success level for the period between 1 February and 30 June 2020 [17,18,44]	Determined by three indicators, namely, (i) cumulative cases, (ii) average daily cases, and (iii) mortality rates	The higher the number of the three indicators (i.e., high cumulative cases, mortality rate, and average daily cases), determined by the abovementioned attributes and interaction activities, the lower the success level of COVID-19 abatement and vice versa.

**Table 3 ijerph-18-01704-t003:** A scoring range and outcome for the Government Stringency Index, the Health and Containment Index, and the Economic Support Index.

Score	Outcome
>90	High
70–89	Medium
<70	Low

**Table 4 ijerph-18-01704-t004:** Values for each rank and the range of score determining the outcome.

Rank	Value	Score	Outcome
First	7	16–21	High
Second	6	10–15	Medium
Third	5	3–9	Low
Fourth	4		
Fifth	3		
Sixth	2		
Seventh	1		

**Table 5 ijerph-18-01704-t005:** A comparative analysis of institutional–social–ecological attributes for explaining the seven countries’ interaction and outcome.

IAD-Based SES Exogenous Attributes	Indonesia	Japan	Malaysia	New Zealand	Singapore	South Korea	Vietnam
**Community/social attributes**							
Low population density	274.2 million (2020) 151 per km^2^ (2020) P	126.5 million (2020) 347 per km^2^ (2020) PP	32.5 million (2020) 99 per km^2^ (2020) P	4.886 million (2020) 18 per km^2^ (2020) P	5.6 million (2020) 8358 per km^2^ (2020) A	51 million (2020) 527 per km^2^ (2020) A	97 million (2020) 314 per km^2^ (2020) PP
High social homogeneity	PP	P	PP	PP	PP	P	P
High level of trust	PP	A	PP	P	P	PP	P
Sufficient local management knowledge and experience	PP	A	P	PP	PP	P	PP
Effective foreign workers influx containment	N/A	N/A	A	N/A	PP	N/A	N/A
**Physical system attributes**							
High facility adequacy	A	P	PP	P	P	P	PP
High technology availability	P	P	PP	PP	P	P	P
High economic Performance	PP	P	PP	P	P	P	A
**Governance/institution attributes**							
Local (top-down) leadership	PP	PP	PP	A	PP	PP	P
Penalty	PP	A	P	P	P	P	P
Strict lockdown	PP	A	P	P	P	PP	P
Strict standard of procedure in public areas	P	PP	PP	P	P	PP	P
Emergency response plan and COVID-19 testing policies	A	PP	PP	P	P	PP	PP
Frequency or co-occurrence of attributes	3Ps 7PPs 2As	4Ps 4PPs 4As	4Ps 8PPs 1A	8Ps 3PPs 1A	8Ps 4PPs 1A	6Ps 5PPs 1A	7Ps 4PPs 1A
**Interaction arena (effectiveness of response/activities)**							
Stringency Index	80 Medium	47 Low	75 Medium	96 High	76 Medium	82 Medium	96 High
Health and Containment Index	69 Low	41 Low	71 Medium	79 Medium	73 Medium	74 Medium	85 Medium
Economic Support Index	25 Low	75 Medium	75 Medium	62 Low	100 High	50 Low	25 Low
**Outcome** **(rank)**							
Cumulative cases	56,385 (7th)	18,600 (5th)	8631 (3rd)	1528 (2nd)	43,894 (6th)	12,839 (4th)	353 (1st)
Average daily cases	466 (7th)	123.2 (5th)	57.2 (3rd)	12.3 (2nd)	290.7 (6th)	85.0 (4th)	2.34 (1st)
Mortality rates	5.1% (6th)	5.2% (7th)	1.4% (3rd)	1.44% (4th)	<0.1% (2nd)	2.2% (5th)	0.0% (1st)
Scores	4	7	15	16	10	11	21
**Success level of COVID-19 abatement**	**Low**	**Low**	**Medium**	**High**	**Medium**	**Medium**	**High**

**Table 6 ijerph-18-01704-t006:** Citizen’s trust towards the government handling the pandemic.

	Indonesia	Japan	Malaysia	New Zealand	Singapore	South Korea	Vietnam
Percentage perceiving government as untruthful [56]	0.486	0.602	0.165	0.024	Between 0.083 to 0.092	0.102	0.029
Public Information Campaign [18]	Coordinated information campaign	Coordinated information campaign	Coordinated information campaign	Coordinated information campaign	Coordinated information campaign	Coordinated information campaign	Coordinated information campaign

**Table 7 ijerph-18-01704-t007:** Countries’ past experiences in handling pandemics.

Experiences in Previous Diseases	Indonesia	Japan	Malaysia	New Zealand	Singapore	South Korea	Vietnam
SARS	Yes	No	Yes	Yes	Yes	Yes	Yes
MERS	No	No	Yes	No	No	Yes	No
H1N1	Yes	Yes	Yes	Yes	Yes	Yes	Yes

Source: [57,62].

**Table 8 ijerph-18-01704-t008:** Components that affect the adequacy of facilities.

	Indonesia	Japan	Malaysia	New Zealand	Singapore	South Korea	Vietnam
Density of physician (per 10,000)	3.777 (2018)	24.118 (2018)	15.132 (2018)	30.252 (2018)	23.063 (2018)	23.661 (2018)	8.199 (2018)
Healthcare facilities (per 10,000)	10.4 beds (2017)	129.8 beds (2018)	18.77 beds (2017)	25.7 beds (2019)	24.86 beds (2017)	124.3 beds (2018)	31.8 beds (2013)

Source: [69,70].

**Table 9 ijerph-18-01704-t009:** Technology availability for COVID-19 abatement.

	Indonesia	Japan	Malaysia	New Zealand	Singapore	South Korea	Vietnam
Contact tracing application	Yes	Yes [76]	Yes [75]	Yes [77]	Yes [31]	Yes [24]	Yes [78,79]
Test kit production	Yes [80,81]	Yes [82]	No	No	Yes [31,83]	Yes [24]	Yes [55]
Ventilator production	Yes [84]	Yes [85]	Yes	Yes	Yes [86]	Yes	Yes [87]
Personal protective equipment (PPE) production	Yes	Yes	Yes	Yes	Yes	Yes	Yes [79]

Source: [18,24,54,75,76,77,78,79,80,81,82,83,84,85,86,87,88,89,90,91].

**Table 10 ijerph-18-01704-t010:** Economic performance of the seven Asia-Pacific countries in 2019.

	Indonesia	Japan	Malaysia	New Zealand	Singapore	South Korea	Vietnam
Economic status	Upper middle-income	High-income	Upper-middle income	High-income	High-income	High-income	Lower middle-income
GNI per capita (ranking based on the Atlas method)	USD 4050 (118^th^)	USD 41,690 (27^th^)	USD 11,200 (68^th^)	USD 42,670 (24^th^)	USD 59,590 (11^th^)	USD 33,720 (30^th^)	USD 2540 (141^th^)

Source: [95,96].

**Table 11 ijerph-18-01704-t011:** Local leadership attributes.

	Indonesia	Japan	Malaysia	New Zealand	Singapore	South Korea	Vietnam
Democracy Index	Flawed democracy	Flawed democracy	Flawed democracy	Full democracy	Flawed democracy	Flawed democracy	Authoritarian

Source: The Economist Intelligence Unit [97].

**Table 12 ijerph-18-01704-t012:** A breakdown of penalties employed during COVID-19.

	Indonesia	Japan	Malaysia	New Zealand	Singapore	South Korea	Vietnam
Fleeing quarantine	1-year/USD 7116.27 [102]	-	2 years imprisonment/fine, or both (1^st^) 5 years imprisonment/fine, or both (repeated) [103]	USD 2873.34/6 months [104]	Maximum USD 7526.05/6 months, or both [105]	Maximum USD 9040.70/maximum 1-year imprisonment with labour [106]	USD 86.51 to USD 4235.31 [107]
Violating lockdown	16 months [102]	-	2 years imprisonment/fine, or both (1^st^) 5 years imprisonment/fine, or both (repeated) [108]	USD 2873.34/6 months [104]	Maximum USD 7526.05/6 months, or both [105]	Maximum USD 9040.70/maximum 1-year imprisonment with labour [106]	USD 432.53 (individual), USD 865.06 (organization)/USD 4325.31 business closure [107]
Failure complying to government personnel	16 months [102]	-	USD 2470.00/2 years, or both [108]	USD 2873.34/6 months [104]	-	Fine of USD 4520.35 [106]	-
Spreading fake news	6 years and maximum of USD 71.16 [109]	-	USD 4919.45 /maximum 6 months. Or both [110]	-	Publish notices on false statements, provide link to government website with official clarifications [111]	5 years or maximum fine of USD 9040.70 [112]	USD 432.53–865.06 [113]
Mask	-	-	-	-	USD 225.78– 752.61 [114]	USD 893.11 [115]	USD 12.95 for not wearing and USD 302.77 if not discarded properly [107]

Currency rate exchange as of 26 January 2021, 16:15 h (GMT +08:00). Source: [102,103,104,105,106,107,108,109,110,111,112,113,114,115]

**Table 13 ijerph-18-01704-t013:** Lockdown enforcement of each country.

	Indonesia	Japan	Malaysia	New Zealand	Singapore	South Korea	Vietnam
School closure	Yes [102]	Yes [53]	Yes [23]	Yes [77]	Yes [116]	Yes [117]	Yes [78]
Border closure	Yes	Yes [53]	Yes [23]	Yes [77]	Yes [21]	Yes [54]	Yes [78]
Gathering bans	Limited to 5 at one time [102]	Yes [53]	Yes [23]	Yes [77]	Yes [21]	Yes [54]	Yes [78]
Work from home	Yes [118]	Yes [76]	Yes, except essential workers [23]	Yes, except essential services [77]	Yes, except essential workers [119]	Yes [117]	Yes
National lockdown	No [93]	No [53]	Yes [23]	Yes [120]	Yes [119]	No [117]	Yes [121]
Localized lockdown	Yes [122]	No [123]	-	-	-	Yes [117]	-
Military enforcement	Yes [124]	No [53]	Yes [125]	Yes	Yes [31]	Only for disinfecting process	Yes [98,126]

Source: [19,23,53,54,55,68,78,94,98,102,117,118,120,121,123,126,127].

**Table 14 ijerph-18-01704-t014:** SOPs in respective countries.

	Indonesia	Japan	Malaysia	New Zealand	Singapore	South Korea	Vietnam
Mask wearing	Compulsory [128]	Habitual [53,129]	Recommendation [130]	Compulsory [131]	Compulsory [119]	Habitual [54]	Compulsory [132]
Social distancing	Yes [133]	Yes [76]	Yes [23]	Yes [77]	Yes [21]	Yes [117]	Yes [55]
Health checks and temperature scanning	Yes [118]	Yes [134]	Yes [23]	Yes [77]	Yes [21,31]	Yes [54]	Yes [68]
Mass sterilization	Yes	Yes [129]	Yes	Yes [135]	Yes	Yes [54]	Yes

Source: [21,23,31,53,54,55,68,76,77,117,118,128,129,130,131,132,133,134,135]

**Table 15 ijerph-18-01704-t015:** Countries’ emergency response plans and COVID-19 testing policies.

	Indonesia	Japan	Malaysia	New Zealand	Singapore	South Korea	Vietnam
Economy stimulus (income support and debt/contract relief)	Covers < 50% of lost salary with narrow relief [18]	Covers < 50% of lost salary with narrow relief [18]	Covers < 50% of lost salary with broad relief [18]	Covers > 50% of lost salary with broad relief [18]	Covers > 50% of lost salary with broad relief [18]	Covers < 50% of lost salary with narrow relief [18]	Covers < 50% of lost salary with narrow relief [18]
Repurposing buildings into healthcare facilities	Yes [84]	Yes [136]	Yes [23]	Yes	Yes [31]	Yes [137]	Yes [55,138]
COVID-19 testing policies	Those with symptoms and key groups only [18]	Anyone with symptoms [18]	Open public testing, including asymptomatic individuals [18]	Anyone with symptoms [18]	Anyone with symptoms [18]	Open public testing, including asymptomatic individuals [18]	Open public testing, including asymptomatic individuals [18]

Source: [19,31,53,55,65,79,94,127,137,138,139,140,141,142].

## Data Availability

The data presented in this study are available in this article.

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
