# Peer review of "Factors Influencing Asia-Pacific Countries’ Success Level in Curbing COVID-19: A Review Using a Social–Ecological System (SES) Framework"

_ijerph, 2021, doi:10.3390/ijerph18041704_

Round 1
Reviewer 1 Report
The topic of the paper is extremely relevant in the time of Covid19 crisis, because it explores what attributes are most significant in curbing the pandemic.
The main concern the paper raises is the sources of the data and the comprehensiveness of the data and the exact comparability of the data:
-line 119 quotes literature review but no exact explanation of the literature considered is given,
-line 165-167 quotes the secondary resources used and makes a list of sources, but it remains generic,
-line 183-184 quotes data were inconsistent or unavailable, that is to say it expresses some problems with the data were found.
Another element to consider is the reliability of the data because different countries have adopted different approaches in declaring their deaths and spread of the virus.
It is absolutely necessary that the author explicits the sources in a systematic way (per country and per kind of data) and recognizes the limits of the research.
Another concern the paper raises is how the 14 attributes were identified; line 164: the meaning of the sentence is not clear.
Given this starting point it can be said this paper is a first overview of possible assessment of the attributes and dimensions that impact the pandemic. The valuable contribution is the putting together of elements, but the findings are rather limited.
line 61: the meaning of the sentence is not clear.
Author Response
Reviewer 1 comments:
1. The topic of the paper is extremely relevant in the time of Covid19 crisis, because it explores what attributes are most significant in curbing the pandemic.
Response: Thank you very much for the encouraging comment.
2. The main concern the paper raises is the sources of the data and the comprehensiveness of the data and the exact comparability of the data:
Response: Thank you for highlight such important concerns. We agreed with the reviewer and there are three issues raised which are (i) sources of data; (ii) comprehensiveness of the data; and lastly exact comparability of the data. To address the first issue, we ensure that those sources of data supporting claims/statement made are primarily derived from scholarly scientific journals/articles as well as legitimate, valid websites including official governmental data. Compared to the original submission, where necessary, we reduced the number of grey literatures (e.g., news except for some international reputable, reliable and accurate ones, e.g., Reuters).
Secondly, in terms of data comprehensiveness, instead of relying only on one source, multiple sources of data were used as part of triangulating strategy to corroborate claims and arguments in the paper.
And lastly, for better comparability of different SES data across seven countries, the same sources of data (e.g., Our World in Data) were used so that although data across seven countries are varied, they can be easily comparable, e.g., in terms of mortality rates (the way how it is defined and measured) as well as the three indices by the Our World in Data developed by the Oxford University. Next, to improve further especially the comparability of SES data, it concerns whether the data collected are truly valid; thus, an operational framework (see Table 2 Page 8) consisting of the list of SES attributes, sub attributes, and their working definitions setting out a clear direction (what data to be collected and sought) is important to ensure data validity/accuracy (quality) which thus enhances data comparability. For example, for the data on population density (i.e., population/land hectares), all seven countries may be measured using the formula.
For the data analysis, the coding process (code assignation) (see Line 229-231 and Page 12) using the following codes based on the criteria: Mostly Present/Present (P), Sometimes/Partially Present (PP), and Mostly Absent/Absent (A), which standardises can help enhance SES data comparability across seven countries. Please also see Table 3 and Table 4 (page 13) for data standardisation that ensure better data or result comparability.
3. line 119 quotes literature review but no exact explanation of the literature considered is given,
Response: Thank you for highlighting the issue. We have further explained what the literature is about. Please see the revised text “Via the above literature review of SES design principles in terms of primary attributes and their sub attributes in relation to the COVID-19 context” on page 4, Line 130-131.
4. line 165-167 quotes the secondary resources used and makes a list of sources, but it remains generic,
Response: Thank you for pointing out the issue of unclearness about the statement. We have revised it by providing more details. Please see page 12 line 202-206 that the revised version is as follows: “This study solely made use of secondary sources, such as research papers, government documents/policies, official newsletters, and websites, specifically pertaining to gathering COVID data based on the predetermined SES attributes across seven countries, statistical data in terms of the number of COVID cases, and COVID-related research findings on factors influencing COVID severity or transmission risks as well as governments’ responses when dealing with the pandemic.”
5. line 183-184 quotes data were inconsistent or unavailable, that is to say it expresses some problems with the data were found.
Response: Thank you for pointing this out. Some data for some countries are indeed are unavailable (e.g., data for the foreign worker containment measures). However, for the issue of inconsistency of data, it does not cause much concern because most of the data based on the sources are consistent. However, when it comes to inconsistency, we resolved by following or using more credible and scientific sources (e.g., governmental official data) (avoiding grey literatures), still, when necessary the latter are applicable to fill the data gap. To avoid confusion, we removed the word inconsistent.
6. Another element to consider is the reliability of the data because different countries have adopted different approaches in declaring their deaths and spread of the virus.
Response: Thank you very much for raising this concern. We apologise for not making this issue. To minimise the issue of reliability data, especially on the definition or interpretation of death and spread of the virus, we only used one established source (i.e., Our World in Data by the University of Oxford) to standardise the meaning and definition of those criteria (in terms of number of death, cumulative cases, as well as daily confirmed cases- 7-day moving average). Also, by this, it increases the comparability of data across seven countries.
In addition, although data were obtained from the same source, in terms of dates of the data, they are varied as displayed in the Our World in Data website; therefore, for data consistency and accuracy, the data were processed (standardised) beforehand prior to using them for the analysis. For example, since the timeframe focused here is from 1st February to 30th June 2020, cases displayed in terms of cumulative cases, daily confirmed cases, and mortality rate prior to the 1st February 2020 were excluded from this study. This is important to reduce data distortion causing imprecision of the result later, especially in computing average daily cases, cumulative cases, and mortality rate across seven countries for the very specific timeframe (see Table 1 Page 5, Line 153, see also Figure 3, 4, and 5, Page 6-7).
To further substantiate reliability or consistency of data, the analysis via the coding process (whether to assign P, PP, or A for each attribute), the inter coder reliability analysis between two coders using the function of Cohen’s Kappa was carried out where the cut-off point of 0.7 and above for substantial reliability/agreement was satisfied (see line 266-267, Page 13).
7. It is absolutely necessary that the author explicits the sources in a systematic way (per country and per kind of data) and recognizes the limits of the research.
Response: Thank you for the valuable and helpful suggestion. Since this study is data driven that necessitates many references or data sources (as evidence) to support the claims, we in this revised version provide sources for each country and attribute in a clearer manner by separating those sources and specifically reference the data where necessary. It is hoped that this version is better compared to the previous or original submission where all sources were cluttered and put under one table. Next, for a better data and result presentation, not only have the sources been accurately provided per country and per kind of data, the referencing style is changed from APA to the Vancouver (numbering) style.
Next, for the limits of the research, we highlighted them in the conclusion and recommendation section (see Page 28, Line 690-697)
8. Another concern the paper raises is how the 14 attributes were identified; line 164: the meaning of the sentence is not clear.
Response: Thank you for highlighting this crucial issue. We apologise as we did not in the first place make it clear in our original submission. For the 13 exogenous attributes, we identified those variables based on (i) the established literature of the SES in commons management (e.g., presence of local leadership, homogeneity level, resources/technological or facility availability); (ii) the mainstream and relevant literature and findings of COVID in terms of potential factors that affect the pandemic; and lastly, sufficient availability of data or information particularly pertaining to factors influencing the pandemic is considered too.
At the same time, for the interaction arena attributes comprising the government stringency index, the health and containment index, and the economic index, they are chosen as they represent activity levels or response effectiveness in line with the SES principle. Meaning that the aforesaid 13 attributes may determine the activities or responses taken by the government whether they are effective or not that eventually influences the pandemic.
Please see the text Line 185-197, page 7-8) where it incorporates the above into the revised paper (see also Table 2 on the operational framework where references are provided).
Lastly, the three indicators (namely mortality rate, cumulative cases, and average daily cases) used to determine outcome in terms of severity or success level of COVID abatement are based on several scholars’ assertion (e.g., Su et al., [13], Dempere [39]- please see page 13 and Line 249-252)
9. Given this starting point it can be said this paper is a first overview of possible assessment of the attributes and dimensions that impact the pandemic. The valuable contribution is the putting together of elements, but the findings are rather limited.
Response: Thank you very much for the encouraging and positive comment on the novelty of this study. Yes, we agree with the reviewer that the primary contribution of this paper is to bridge the SES knowledge with COVID-19 by providing a first overview of SES influencing factors towards explaining the COVID-19 severity or abatement level. Aside from affirming the significance of SES factors/attributes co-occurrence/presence (in terms of number of frequency) in the COVID context, the findings also provide what and how (or to what extent) SES factors, and also which factors appear to be more significant to, influence the pandemic outcome. In short, although quite limited, the findings offer interesting patterns of SES factors towards the possibilities for controlling the growth of the pandemic.
At the same time, since this explorative study is among the first, conceptualising the SES in the COVID setting, we also highlighted limitations in the conclusion section where due to the limited sample size (i.e., only 7 countries are taken as cases), some inferential statistical analyses (e.g., regressional evidence to clearly show significant associations between those variables and the outcome) may be quite limited. Nevertheless, as mentioned, despite this limitation, we believe that the paper at this point of time offering insightful findings can be useful for the future reference (direction), particularly for more effective policymakers’ decision making (see the conclusion section, Page 27).
10. line 61: the meaning of the sentence is not clear
Response: Thank you for pointing out the unclearness issue about the sentence. What we intend to posit here is that, since the SES framework, synonymous with commons and resource management or collective action, is rarely applied in a health and disease-related topic, it is necessary to be explored, especially in the context of COVID-19. Therefore, to make the meaning clearer, we modified the sentence to “relationships between the transmission risk or severity of COVID-19 and socio-economic and environmental factors via the SES lens should be explored in order to devise a holistic strategy in mitigating the pandemic” (see Page 2, Line 66-69).

Reviewer 2 Report
The paper by Ling et al. “Factors influencing Asia-pacific countries’ success level in curbing covid-19: a review using a social-ecological system framework” presented a social-ecological framework to assess the effectiveness of different factors (population, technology, etc) on curbing covid-19. L13 SES should be explained first. L53 IAD should be explained first. Figure 3 What are the grey lines? Y-axis seems to be log scaled. This should be mentioned in the caption. 117 seems to be the smallest y-intercept. This should be explained. Figure 4 The colors of Indonesia and Singapore need to be more distinguishable. L166-167 Need to use other common data sources, such as Web of Science, PubMed. A lot of reference sources are news. They are not the usual types of academic papers (with peer review). L183 mentions the coding is also very intuitive and subjective. I am afraid this makes the paper much less convincing.
Author Response
Reviewer 2 comments:
1. The paper by Ling et al. “Factors influencing Asia-pacific countries’ success level in curbing covid-19: a review using a social-ecological system framework” presented a social-ecological framework to assess the effectiveness of different factors (population, technology, etc) on curbing covid-19.
Response: Thank you for encapsulating well the intent of this study.
2. L13 SES should be explained first.
Response: Thank you for highlighting this as the previous version was not clear. We provided a full term of SES in the abstract section; however, more detailed definitions are embedded in the text (see line 84-86, page 2). The added in text is as follows “The SES framework provides a more detailed variable oriented analysis of the social-ecological system (Cole et al., 2019). According to Partelow (2018), SES is “a conceptual framework providing a list of variables that may be interacting and affecting outcomes in social-ecological systems”. Via the identified SES attributes, consistent with the application spirit of IAD, they are able to diagnose and explain the interaction (activities) and the outcome of a situation”
3. L53 IAD should be explained first.
Response: Thank you for the suggestion. For better story line building up and continuity, we added in a clearer definition or explanation in the introduction under a specific section (see Line 71-80, page 2). The revised text reads as follows: “…is primarily used to evaluate the effects of institutional arrangements, as well as the physical environment and local community in explaining a certain contextual outcome. It was envisioned as a systematic tool for scholars of different disciples to communicate with each other regardless of their broad perspectives to pave a way towards better understanding of a situation (Cole et al., 2019)[7]. Within the framework, it is important to understand the action situation/interaction that actors are in, plausible choices made by them and how it will affect the pattern of the outcome. In order to predict choices that will be made, it is crucial to know: (i) the resources brought into the situation, (ii) the valuation assigned by the actor to the state of the world and to the action; (iii) how the actors acquire, process, retain and use the knowledge and information; and (iv) ways used by the actor in selecting a particular decision (Mcginnis et al., 1992)[8]”
4. Figure 3 What are the grey lines? Y-axis seems to be log scaled. This should be mentioned in the caption. 117 seems to be the smallest y-intercept. This should be explained.
Response: Thank you for the question and suggestions. We have modified the graph to a better version which shows clearer all the lines for the countries and the graph is in the linear form. Please see Page 6, Line 158, and Figure 3.
5. Figure 4 The colors of Indonesia and Singapore need to be more distinguishable.
Response: Thank you for the suggestion. we have edited the graph to a better version which shows distinctive colours between Indonesia and Singapore. Please see page 6 and Figure 4.
6. L166-167 Need to use other common data sources, such as Web of Science, PubMed. A lot of reference sources are news. They are not the usual types of academic papers (with peer review).
Response: Thank you for raising this important issue. We totally agree with the reviewer that it is vital to provide more academic data sources as they represent credibility. Therefore, to improve this situation, we removed where necessary the grey literature including news and added in more reliable and credible academic literature to support the results and findings of this study. However, we have retained some of the news data since academic data or literature may not be unavailable.
7. L183 mentions the coding is also very intuitive and subjective. I am afraid this makes the paper much less convincing.
Response: Thank you for pointing this out. We have modified the sentence. Although the code assignation process involving human input which is inherently biased, we try to keep biasness and inconsistency at the most minimal level in order to make the qualitative findings more credible/valid and trustworthy by providing an operational framework consisting of variables/attributes and working definitions, a guide on how to assign code for each attribute (e.g., when to assign P, PP, or A), and lastly a reliability test via Cohen’s Kappa analysis. The test was carried out to ensure the consistency of the coding process between two coders. With all these measures taken, it is hoped that the analysis becomes more robust, systematic and scientific. Please refer to Page 12-13, Line 225-260 in the revised paper on the approaches and techniques used for the data analysis.

Reviewer 3 Report
The main objective of this paper is to assess the relationship between governmental measures and the success level in tackling the expansion of COVID-19 infection. To do so the authors focus their study in seven Asian-pacific countries. In the manuscript the authors describe their method for evaluating the success level of governmental measures like: “past experiences facing similar diseases, facilities 21 mobility, lockdown measures, imposition of penalty, and standard of procedures in public spaces are 22 deemed significant in determining the severity of a country”. The study achieves a seven country rank indicating an interesting patterns toward the possibilities for controlling the growth of the pandemic.
General Comments
In some sections the analysis is based, more than in real evidence, on speculations based on circumstantial data. In Section regarding “Management Knowledge and Experience” for example, inferences are set upon previous experiences of diseases in several countries. The way these paragraphs are written are too close to ensuring that having gone thru a contagious disease, implies getting knowledge about dealing with these situations. The authors may, just ‘softening” a bit the writing, improve the precision of this manuscript.
The Emergency Response Plans are the same for all countries. This parameter does not provide information to regard differences among countries.
The manuscript describes the source of the data included in this study. But a mayor failure resides in the fact no methodology to make an actual analysis is shown in this study. An important drawback is this study exhibits a good selection and retrieval of data, but the actual analysis of the data limits to make some qualitative, therefore unprecise, comments. The variables describing the present scenario in the seven countries studied are treated in an independent fashion. All over the document the authors express the little presence every single factor has in explaining the pandemic experience for each country. I feel this study has collected interesting data. However, the crucial use of the data to produce useful results is not present.
Writing and organization
This paper is grammatically well written.
The Section Study Areas presents a precise narrative of the COVID-19 starting process in the studied countries. Very nice writing that keeps the reader’s interest in the document.
Line 214: “In contrast, Indonesia fumbled despite 214 having a lower population density due to the shortage of facilities and economy” How do you know this? Please leave some other possibility including a “seem” or some indication of likelihood as “presummbaly”into the sentence.
Minor Details
Line 129: The cumulative confirmed cases for Vietnam Is not consistent with the number in Table 1
The term increase” used to qualify attributes of the process as in Figure 4 for Daily average case Increase, is confusing. If it were an actual increase, this value should get negative values in some stages of the process. Am I wright? Or is there another possibility?
Table 3 is very important since it summarizes the results. It must be properly aligned, otherwise is not readable. Perhaps setting the dimensions only once in the first column helps in the sense of using better the space.
Table 9 refers to different currencies making it difficult to interpret the data. Please add a reference unique currency.
Writing numbers which are lower than ten using letters (not numerical digits) is probably better adapted to usual writings codes.
Final Statement.
The authors have collected an important amount of data describing the experience of several Asian-Pacific countries. To make this study publishable a better, more precise method to compare among countries experience, should be developed. Just commenting about the data is not enough.
Contrasting the impact of the factors you study one versus each other would be a way to substantially improve this paper.
Author Response
Reviewer 3:
1) The main objective of this paper is to assess the relationship between governmental measures and the success level in tackling the expansion of COVID-19 infection. To do so the authors focus their study in seven Asian-pacific countries. In the manuscript the authors describe their method for evaluating the success level of governmental measures like: “past experiences facing similar diseases, facilities 21 mobility, lockdown measures, imposition of penalty, and standard of procedures in public spaces are 22 deemed significant in determining the severity of a country”. The study achieves a seven country rank indicating an interesting patterns toward the possibilities for controlling the growth of the pandemic
Response: Thank you very much for the comments and summarisation of this study. This is an accurate depiction of this paper’s intention.
General Comments
2) In some sections the analysis is based, more than in real evidence, on speculations based on circumstantial data. In Section regarding “Management Knowledge and Experience” for example, inferences are set upon previous experiences of diseases in several countries. The way these paragraphs are written are too close to ensuring that having gone thru a contagious disease, implies getting knowledge about dealing with these situations. The authors may, just ‘softening” a bit the writing, improve the precision of this manuscript.
Response: Thank you noting this flaw in this paper. Indeed, we do not intend to overclaim our statement, as if it entails a causal-effect link for this section. We agree that the statement appears as if for one to gain knowledge in managing a pandemic, one must experience or go through a contagious disease. This claim on the direct association may not always be true. So, we heed the reviewers’ advice to revisit this section and other sections to carefully check and ‘soften’ the writing in order to improve the precision of meanings in this paper (see Line 356-378, Page 17-18).
3) The Emergency Response Plans are the same for all countries. This parameter does not provide information to regard differences among countries.
Response: Thank you for the comment. We carefully checked back, added another point which is on COVID-19 testing policy, and recoded or assigned a different code for some countries, based on the sub attributes; now there are variations or differences among countries. Please see Table 5, page 15 and See Table 15, page 25.
4) The manuscript describes the source of the data included in this study. But a major failure resides in the fact no methodology to make an actual analysis is shown in this study. An important drawback is this study exhibits a good selection and retrieval of data, but the actual analysis of the data limits to make some qualitative, therefore unprecise, comments. I feel this study has collected interesting data. However, the crucial use of the data to produce useful results is not present
Response: Thank you for highlighting the weakness of the data analysis in the original manuscript. We apologise for the unclearness of this paper’s data analysis methodology. In the revised version, we improved the issue where with the proposed operational framework in Table 2 consisting of the list of SES attributes, sub attributes, and their working definitions setting out a clear direction (what data to be collected and sought) to ensure data validity (quality), it helps enhance the analysis (i.e., the qualitative coding process). In short, the data analysis methods primarily used in this study were the qualitative content (textual based) analysis (which is a descriptive one), via a coding system (with P, PP, and A) based on the criteria set, i.e., ratio/level or percentage of DP presence/occurrence, i.e., Mostly Absent/Absent (A) (with 0%-29%), Sometimes Present/Partially Present (PP) (with 30%-69%) or Mostly Present/Present (P) (with 70%-100%), supplemented by the frequency analysis in terms of co-occurrence of those codes, and the scoring system for the ranking analysis. For the reliability or consistency level between two coders, the inter-rater reliability of Cohen’s Kappa using SPSS was conducted. Although it was predominately analysed in a qualitative fashion, it is corroborated by some simple quantitative statistical analyses. With more systematic analyses, it is hoped that it adds precision and accuracy to the findings. Please see Line 225-255 and Page 12-13, where the revised version of text is as follows:
“The study primarily employed a qualitative content analysis to study institutional-social-ecological attributes, using Ostrom’s eight design principles [6] as one of the theoretical underpinnings. To aid the comparative analysis, a coding system, based on both textual and quantitative evidence, was constructed and used to determine IAD based SES attribute occurrence in each country. Meaning that, each attribute was assigned with a code based on the ratio/level or percentage of DP/attribute presence/occurrence [40], i.e., Mostly Absent/Absent (A) (with 0%-29%), Sometimes/Partially Present (PP) (with 30%-69%) or Mostly Present or Present (P) (with 70%-100%). This coding process, particularly for the comparative analysis of a limited number of cases where a statistical inferential analysis is not possible, is scientifically appropriate and relevant since it is based on several SES scholars’ systematic methodologies which have been widely applied in different resource management contexts [41–43]. Ultimately, in line with Ostrom’s successful commons governance focusing on the importance of the presence of design principles, a higher frequency/co-occurrence of P indicates a better COVID-19 containment, and therefore a higher success rate of the country, while a lower count of P connoting a higher frequency of A or PP means otherwise. Besides, a quantitative method, apart from providing the ratio/percentage of DP occurrence, assessing the frequency of attributes presence as well as for the purpose of calculating scores and assigning ranks was also used to supplement the qualitative analysis. To ensure the consistency or credibility of the coding process between two coders, inter-rater reliability of Cohen’s Kappa using SPSS was conducted for each attribute under the institutional-social-ecological factors, and finally the overall average reliability for all 12 SES attributes (excluding effective foreign worker containment) across seven countries has also been obtained. The cut-off point for substantial reliability/agreement is 0.7 and above [44,45]. Next, the three indices which form the interaction arena (government responses/activities), based on the scores, were assigned with different codes, namely High (H), Medium (M) and Low (L) (see Table 3).
At the same time, the success level of a country (outcome), influenced by the above DPs and three indices, was measured and determined based on the total scores obtained from the three criteria, namely the cumulative number of cases of each country, average daily confirmed cases, and the mortality rate of the country [13,39]. More precisely, the total scores were calculated based on the ranking among the 7 countries (i.e., 1st to 7th) and values assigned (from 1 to 7) to each criterion. Table 4 summarizes the options of values for each rank as well as the score ranges (from 3 to 21) for the high, medium, and low outcomes, respectively.”
5) The variables describing the present scenario in the seven countries studied are treated in an independent fashion
Response: Thank you for highlighting this. Based on the SES framework spirit, the three primary institutional-social-ecological attributes are deemed to be exogenous or those variables are independent. To make the framework clearer, we also added in the text that those variables are exogenous, however, they are discussed in a more interconnectedly fashion in the action or interaction arena (i.e., responses or activities) that leads to the outcome (i.e., success level of COVID-19 abatement). More precisely, Table 5 (Page 14) (the summary of the result) is the evidence and method/approach used by scholars, showcasing how those exogenous attributes eventually come together to explain or justify the activities and responses taken in the interaction arena as well as the outcome. Perhaps, the current presentation fashion showing independent effects of attributes towards the outcome becomes even more prominent because after the summary result, we discuss each attribute/variable (what data or current scenario) across seven countries and its effect individually towards influencing or determining the final outcome. It is necessary to be carried out this way because it shows a better picture by further discussing how every case/country’s each SES attribute impacts its COVID-19 outcome.
6) All over the document the authors express the little presence every single factor has in explaining the pandemic experience for each country.
Response: Thank you for the comment about the need to clarify the impact of SES attributes towards explaining the pandemic abatement level. We hope that we understand the reviewer’s intention of this comment. We apologise that in the original submission this was not clearly discussed. Therefore, in this revised version, we have provided an operational framework (see Table 2, Page 8-12) consisting of the list of SES attributes, sub attributes, and their working definitions as well as assumptions that set out a clear direction of the effects/influence of those SES attributes towards pandemic abatement. Not only the effects of each SES attribute/factor are theoretically highlighted in Table 2, each detailed SES attribute is also discussed in the result and discussion section (See from Page 13-27).
7) Writing and organization
This paper is grammatically well written.
Response: Thank you very much for the comment. We also tried to carefully proofread and check the overall text.
8) The Section Study Areas presents a precise narrative of the COVID-19 starting process in the studied countries. Very nice writing that keeps the reader’s interest in the document.
Response: Thank you very much for the comment and encouragement.
9) Line 214: “In contrast, Indonesia fumbled despite 214 having a lower population density due to the shortage of facilities and economy” How do you know this? Please leave some other possibility including a “seem” or some indication of likelihood as “presummbaly” into the sentence
Response: Thank you very much for pointing inaccurate statement. We have modified the statement where we follow the reviewer’s suggestion to use more soft or associative word rather than establishing any causal-effect link here. The revised statement is “In contrast, Indonesia fumbled despite having a lower population density; this is probably due to other SES factors, such as shortage of healthcare facilities, weak institutional arrangement for strict enforcement, and also an economic condition is one of the considerations” see Line 302-304, Page 16 in the revised paper.
Minor Details
10) Line 129: The cumulative confirmed cases for Vietnam Is not consistent with the number in Table 1
Response: Thank you for pointing out the error. We have rectified accordingly. Please see Table 1, Page 5.
11) The term increase” used to qualify attributes of the process as in Figure 4 for Daily average case Increase, is confusing. If it were an actual increase, this value should get negative values in some stages of the process. Am I wright? Or is there another possibility?
Response: Thank you very much for pointing out this issue. To avoid confusion, we used Average Daily Confirmed Cases, instead of daily case increase. Meaning that for the period between 1st Feb and 30th June, we are interested to know how many confirmed cases daily, by average, across seven countries. Please see Table 1, Page 5. See also the explanation from Line 137-144, Page 5.
12) Table 3 is very important since it summarizes the results. It must be properly aligned, otherwise is not readable. Perhaps setting the dimensions only once in the first column helps in the sense of using better the space.
Response: Thank you for your constructive suggestion. This table showing the overall result is indeed a very important one. To improve the quality so as to make it more presentable and understandable, we re-aligned some texts and provided internal borders. Please see Table 5 Page 14-15.
13) Table 9 refers to different currencies making it difficult to interpret the data. Please add a reference unique currency.
Response: Thank you very much for your suggestion. We amended accordingly in the revised paper, where for familiarity and uniformity purposes that help in result interpretation, we used USD as the reference currency. Please see Table 12, Page 22.
14) Writing numbers which are lower than ten using letters (not numerical digits) is probably better adapted to usual writings codes.
Response: Thank you very much for your suggestion. We have followed it and amended accordingly in the revised paper.
Final Statement.
15) The authors have collected an important amount of data describing the experience of several Asian-Pacific countries. To make this study publishable a better, more precise method to compare among countries experience, should be developed. Just commenting about the data is not enough.
Response: Thank you for this very important suggestion. This comment dealing with a methodology in analysing the data is similar with comment no 4 by the reviewer. Therefore, aside from commenting on the data, our response is that we have a set of methodology for the purpose of analysing the attributes, their interactions towards the pandemic outcome (especially using the coding system, based on its specific criteria in terms of code assignation). By this, we hope that once all the 13 SES attributes/factors (see the result in Table 5 Page 14-15) and their sub attributes (in the form of more detailed result of each SES attribute, see Page 16-27) as well as indices within the interactions arena and the outcome across seven countries are standardised using the codes, ranking analysis, and similar sources, they are indeed comparable, and this leads us to making a more valid claims and arguments in the paper.
16) Contrasting the impact of the factors you study one versus each other would be a way to substantially improve this paper.
Response: Thank you for the helpful suggestion to improve the paper. We have constructed a table (see Table 2 Page 8-12) comprising attributes (altogether 13 of them) and their assumptions of those factors’ impacts towards influencing the government responses (interaction arena) and finally the outcome which is the COVID abatement level. Meaning, we can see distinctively each institutional-social-ecological attribute’s or factor’s impact in explaining the outcome. Also, based on the SES conceptual framework, laid as a basis, we used and triangulated them for the 7 country COVID cases analysis, in which each attribute/factor’ impact in terms of what and how is discussed. See also the Result and Discussion section for result comparison across seven countries.

Round 2
Reviewer 1 Report
The comments have been considered, some answers were given and the paper changed accordingly.
Reviewer 2 Report
The authors addressed my concerns. But the reference section needs a overhaul. A lot not in standard format, or missing doi links.
Reviewer 3 Report
A deep writing and formatting effort is notorious. The manuscript now looks much better and almost ready to publish. Graph's scales make them now easier to read.
I would include the reference (fifth) column of Table 2 into the second column (Attributes). This will give more space for columns three and four and make the table more readable. In Tables 12, 15, and perhaps others too, aligning texts to the left may make them look better... just a possibility you may try.
Good job!